# Effect of Ultraviolet Illumination on the Fixation of Silver Ions on Zinc Oxide Films and Their Photocatalytic Efficiency

Dobrina Ivanova [1], Ralitsa Mladenova [2], Hristo Kolev [2] and Nina Kaneva [1,*]

[1] Laboratory of Nanoparticle Science and Technology, Department of General and Inorganic Chemistry, Faculty of Chemistry and Pharmacy, University of Sofia, 1 James Bourchier Blvd., 1164 Sofia, Bulgaria; dobrina.k.ivanova@gmail.com
[2] Institute of Catalysis, Bulgarian Academy of Sciences, Acad. G. Bonchev St., bl. 11, 1113 Sofia, Bulgaria; ralitsa@ic.bas.bg (R.M.); hgkolev@gmail.com (H.K.)
* Correspondence: nina_k@abv.bg

**Abstract:** This study focuses on the fabrication and characterization of nanostructured zinc oxide films deposited on glass substrates using sol–gel dip-coating methods. The thin films are functionalized with silver ions at various $Ag^+$ concentrations ($10^{-2}$, $10^{-3}$, $10^{-4}$ M) through room temperature ion fixation process with and without ultraviolet (UV) illumination. Physicochemical characterization techniques, such as employing Scanning Electron Microscopy with Energy-dispersive X-ray spectroscopy (SEM-EDX), X-ray Diffraction (XRD), X-ray Photoelectron Spectroscopy (XPS), Ultraviolet–Visible Spectroscopy and Electron Paramagnetic Resonance (EPR) techniques. The SEM-EDX and XRD confirmed a characteristic ganglia-like structure with a hexagonal crystalline structure. The photocatalytic performance and available surface area of the pure and Ag films are investigated in the removal of methylene blue dye under UV and visible light illumination and in darkness. It is observed that the photocatalytic activity increases proportionally to the $Ag^+$ ion concentration: $ZnO < Ag(10^{-4} M)/ZnO, < Ag(10^{-3} M)/ZnO < Ag(10^{-2} M)/ZnO$. Moreover, the catalysts modified under UV illumination during the fixation treatment (Ag-UV/ZnO) exhibited a higher photocatalytic efficiency and degraded the dye in comparison with those without a light source (Ag/ZnO). The experimental results are confirmed using total organic carbon (TOC) analysis. The optimal silver concentration ($10^{-2}$ M) is established, which shows the highest photocatalytic efficiency (in both cases of ion fixation treatment). The results can be used as a guideline for the development of co-catalyst-functionalized semiconductor photocatalysts.

**Keywords:** ZnO; sol–gel; silver ions; fixation; photofixation; UV illumination; photocatalysis





## 1. Introduction

Wastewater often contains organic pollutants that may pose a significant threat to living organisms, including humans and animals [1]. The presence of non-degradable dyes in rivers, lakes, and other water sources, has a detrimental impact on the environment [2,3]. Industrial enterprises, being the primary producers of dye waste, contribute substantially to this issue. Among the problematic dyes, methylene blue stands out due to its widespread applications in various sectors such as printing, textiles, and even medical fields like tumor therapy in pharmaceuticals [4,5]. Methylene blue is utilized in medicine to treat conditions like vasoplegic syndrome, congestive heart failure, sepsis, renal failure, hepatic failure, etc. [6,7]. The push to eliminate organic pollutants is gaining momentum globally, driven by the objective of ensuring access to safe and clean drinking water. Addressing the challenges caused by dye discharge of dyes into the environment necessitates the development of novel and effective purification methods.

Various pollutant remediation methods are available for treating dye wastewater, including biological, physical, and chemical processes [8]. Among these, photocatalysis stands out as a promising and effective approach, utilizing a semiconductor material as a

photocatalyst to accelerate the degradation of organic pollutants [9]. Photocatalysts offer the advantage of enhancing the efficiency of the wastewater treatment process by allowing both spontaneous and non-spontaneous reactions to take place [10] and effectively degrade organic dyes into simpler products that pose lower environmental risks. Among the popular photocatalysts, nanostructured semiconductors, such as $TiO_2$, ZnO and $Fe_2O_3$, are preferred due to their availability, cost-effectiveness, and environmental compatibility [11]. Among these semiconductors, ZnO has attracted significant attention owing to its high stability, large surface area, non-toxic nature, and ease of preparation [12,13].

The efficiency in heterogeneous semiconductor photocatalysis is strongly influenced by the lifetime of the photogenerated electron/hole charges ($e^-/h^+$). The recombination of these electron/hole pairs can significantly limit the photocatalyst's activity. To mitigate the recombination process, the semiconductor surface can be modified through functionalization with metallic or non-metallic nanostructures [14]. Generally, noble metals, such as gold, platinum, and silver, are widely used as co-catalysts to improve the photocatalytic characteristics of zinc oxide [13]. Among these, silver stands out as one of the most employed due to its excellent electrical conductivity, chemical stability and high reduction potential. When introduced onto the ZnO surface, Ag nanoparticles can act as electron traps or electron acceptors. Consequently, the lifetime of the electron-hole pairs is extended, leading to an increase in the catalytic efficiency of the semiconductor is increased [15].

With the objective of investigating this co-catalyst modification strategy, this study focuses on evaluating the photocatalytic efficiency of silver co-catalyst zinc oxide (ZnO) deposited via sol–gel dip-coating technique on a glass substrate. A novel aspect of this investigation is to elucidate the impact of ultraviolet illumination, as an aid, for the photofixation of the silver ions on the ZnO films' surface. In the experiments presented herein, we have prepared three types of samples: pure ZnO, ZnO obtained by impregnation–fixation of Ag ions without UV illumination (Ag/ZnO), and Ag/ZnO prepared by UV-assisted photo-fixation (Ag-UV/ZnO). In all cases, the fixation was carried at a varied Ag ion concentration ($10^{-2}$–$10^{-4}$ M range). The photocatalytic experiments were conducted using methylene blue dye as a model contaminant and were also carried out under UV and visible illumination, as well as in darkness.

## 2. Results and Discussion

### 2.1. Characterization of Photocatalysts

The scanning electron microscopy (SEM) images in Figure 1 illustrate the ganglia-like structure of both pure and modified zinc oxide samples, revealing a homogeneous surface morphology. Consistent with our previous results, we observed a reduction in the particle size of ZnO when $Ag^+$ ions were introduced to modify the ZnO crystalline structure [16,17].

Additionally, silver nanoparticles are observed on the surface of the ZnO sample co-catalytically via photo-fixation (Ag-UV/ZnO), predominantly surrounding the ganglia-like structures. Consequently, these silver photo-fixed films are expected to exhibit enhanced photocatalytic efficiency compared to pure zinc oxide.

The elemental composition of Ag-UV/ZnO films is investigated employing Energy-dispersive X-ray spectroscopy (EDX), as shown in Figure 1c, accounting for approx. 3 wt.%. This can be attributed to the small quantities of co-catalyst introduced during functionalization. The presence of Si was most likely due to the glass substrates on which the nanostructured materials are deposited.

The structural properties of pure ZnO and Ag, UV/ZnO films are analyzed using X-ray diffraction (XRD) analysis. Figure 2 displays eight diffraction peaks of ZnO observed 2θ of 31.94°, 34.67°, 36.51°, 48.23°, 56.84°, 63.22°, 67.53°, and 68.18°. These peaks correspond to the lattice plane orientations of each peak are (100), (002), (101), (102), (110), (103), (112), and (201), respectively, which correspond to hexagonal wurtzite structure (JCPDS card No. 96-230-0117) [18]. The diffraction peaks in the pure films suggest showing a crystalline structure without impurities or the presence of other crystalline phases. The XRD patterns of both pure ZnO and Ag-UV/ZnO samples demonstrate sharp peaks, indicating a high

degree of crystallinity [19]. Furthermore, the diffraction pattern of Ag-UV/ZnO, modified at the highest $Ag^+$ concentration $10^{-2}$ M modified zinc oxide, showed additional peaks, located at 2θ angles of 38.46°, 45.86° and 64.72°, which correspond to the metallic form of Ag (JCPDS 96-901-3048) and the planes (111), (200) and (220), respectively. Comparatively, the XRD patterns of pure ZnO-and Ag/ZnO-modified films show no noticeable differences due to the silver ion modification [20].

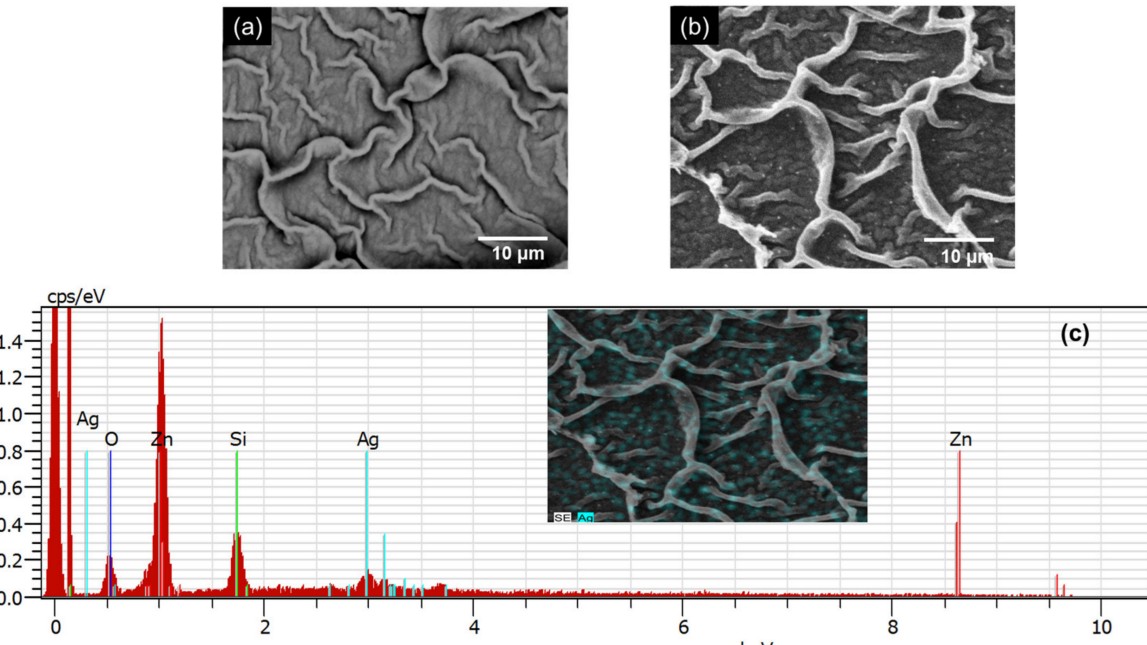

**Figure 1.** SEM images of ZnO (**a**) and Ag-UV/ZnO (**b**) films. EDX spectrum of Ag/ZnO (**c**).

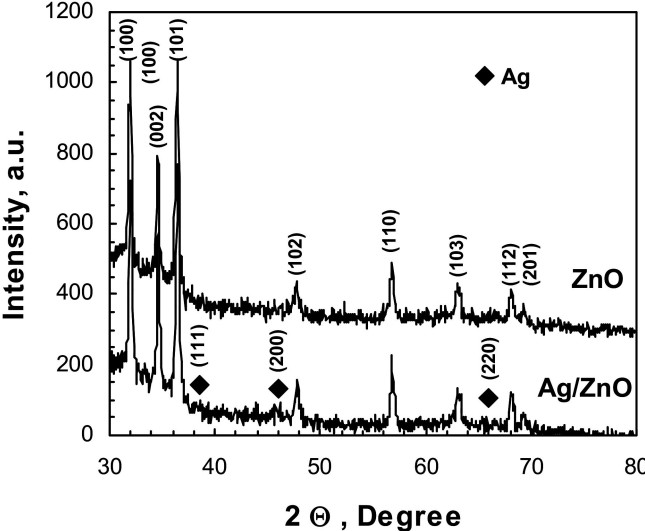

**Figure 2.** XRD patterns of pure and silver co-catalytic-modified semiconductor films.

The crystallite size of both the pure and modified films is calculated using Scherrer equation, which utilizes the Bragg equation to determine the interplanar distance and crystal lattice dimensions. The introduction of $Ag^+$ ions to ZnO did not have a significant impact on the crystallite size. However, the intensity of ZnO peaks decreased upon modification of the film. The average crystallite size showed a slight decrease, with d(ZnO) = 48.4 nm and d(Ag, UV/ZnO) = 42.9 nm. The calculated lattice parameters of Ag-UV/ZnO films (a = b = 3.2518 Å, c = 5.2105 Å) are found to be very close to those of pure ZnO (a = b = 3.2524 Å, c = 5.2124 Å), indicating that the films are maintained the hexagonal

wurtzite structure [21]. Additionally, the microstrain of the films is determined using the *c*-axis lattice parameter. The calculation revealed a positive value indicating tensile strain. The microstrain in Ag/ZnO films ($0.8 \times 10^{-3}$ a.u.) exhibited a slight magnitude of tensile strain compared to pure ZnO ($0.9 \times 10^{-3}$ a.u.).

X-ray Photoelectron Spectroscopy (XPS) is employed to investigate the chemical composition of the prepared materials. The oxidation states of silver co-catalysts and the surface concentration of the elements in the Ag/ZnO and Ag-UV/ZnO films are determined and presented in Figure 3.

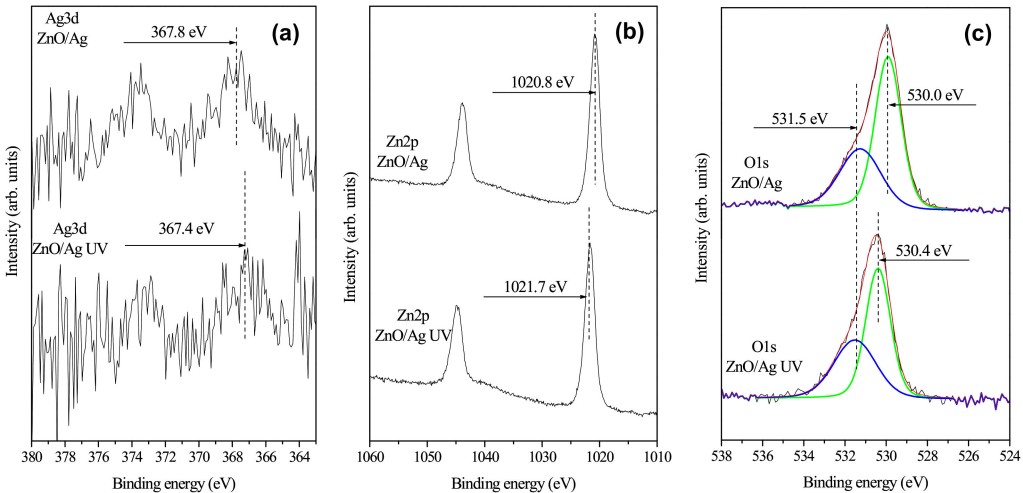

**Figure 3.** High-resolution XP spectra of Ag-modified zinc oxide films: (**a**) Ag3d, (**b**) Zn2p, and (**c**) O1s regions.

Figure 3 depicts the high-resolution XPS spectra of studied silver co-catalysts, namely ZnO, modified via photo-fixation (Ag-UV/ZnO) and ion fixation in darkness (Ag/ZnO). The measured spectra in Figure 3a correspond to the Ag3d region. The observed binding energies (BEs) of approx. 367.8 eV (Ag/ZnO) and 367.4 eV (Ag-UV/ZnO) are consistent with the typical BEs measured for $Ag^+$ ions [12,22–24].

The difference of 0.4 eV can be attributed to the presence of a very low concentration of $Ag^+$, resulting in noisy curves. Figure 3b provides a comparison of the measured Zn2p energy ranges measured for both samples. The measured BEs are 1020.8 eV (Ag/ZnO) and 1021.7 eV (Ag-UV/ZnO), respectively, indicating the presence of the +2 oxidation state of zinc in both samples, in accordance with ZnO [12,23]. The Zn2p spectra exhibit high quality. Therefore, an alternative explanation is required for the BEs discrepancy. Upon closer examination of Figure 3a,b, it becomes apparent that the BE of $Ag3d5/_2$ is higher in the Ag/ZnO sample compared to the BE of Ag3d $5/_2$ in the Ag-UV/ZnO sample. Similarly, the BE of $Zn2p3/_2$ is lower in the Ag/ZnO sample compared to the BE of $Zn2p3/_2$ in the Ag-UV/ZnO sample. Moreover, a similar trend is observed for the O1s peak. In both samples, the O1s peak has been deconvoluted to two subpeaks representing Zn-O bonds with BEs of 530.0 eV and 530.4 eV for Ag/ZnO and Ag-UV/ZnO, respectively, and oxygen vacancies with a BE of approx. 531.5 eV [12,23]. Remarkably, the difference in BE for Zn-O bonds is 0.4 eV, but in the opposite direction. When the Ag3d exhibited a 0.4 eV higher value, the O1s peak showed a 0.4 eV lower value and vice versa.

This behavior could be attributed to different charge transfer dynamics between $Ag^+$ ions and ZnO thin film. The interaction and charge transfer are stronger in the Ag-UV/ZnO sampled compared to Ag/ZnO. Upon irradiation with UV light, the interaction and charge transfer become even more pronounced. Thus, the recombination of the $e^-/h^+$ pairs is reduced. Therefore, Ag-UV/ZnO samples are expected to have higher photocatalytic efficiency compared to those produced via impregnation–ion fixation without UV illumination aid.

The calculated surface atomic concentrations are presented in Table 1. For both samples, the Zn:O ratio is approx. 1:1, which closely aligns with the theoretically expected ratio for ZnO. As anticipated, the silver concentration is negligible due to the small quantity of surface-bound $Ag^+$.

**Table 1.** Surface atomic concentrations of elements presented of Ag-modified zinc oxide films, at. %.

| Sample | O1s | Zn2p | Ag3d |
|---|---|---|---|
| Ag/ZnO | 48.36 | 51.57 | 0.07 |
| Ag-UV/ZnO | 49.51 | 50.41 | 0.08 |

Electron paramagnetic resonance (EPR) spectroscopy was employed to investigate the impact of UV irradiation on defective states and Ag incorporation into the Ag/ZnO samples, EPR spectroscopy is used. The spectra obtained from the Ag/ZnO and Ag-UV/ZnO samples are depicted in Figure 4. All Ag/ZnO films exhibited a prominent EPR line at g = 1.96, indicative of a shallow effective mass donor center in zinc oxide [25]. In the literature, authors connected the shallow donor center has been linked to bulk defects originating from negatively charged Zn vacancies [26,27]. Additionally, a weak signal was recorded at g = 2.002 and is attributed to surface defects originating from singly ionized oxygen vacancies ($V_0^+$) [25,27].

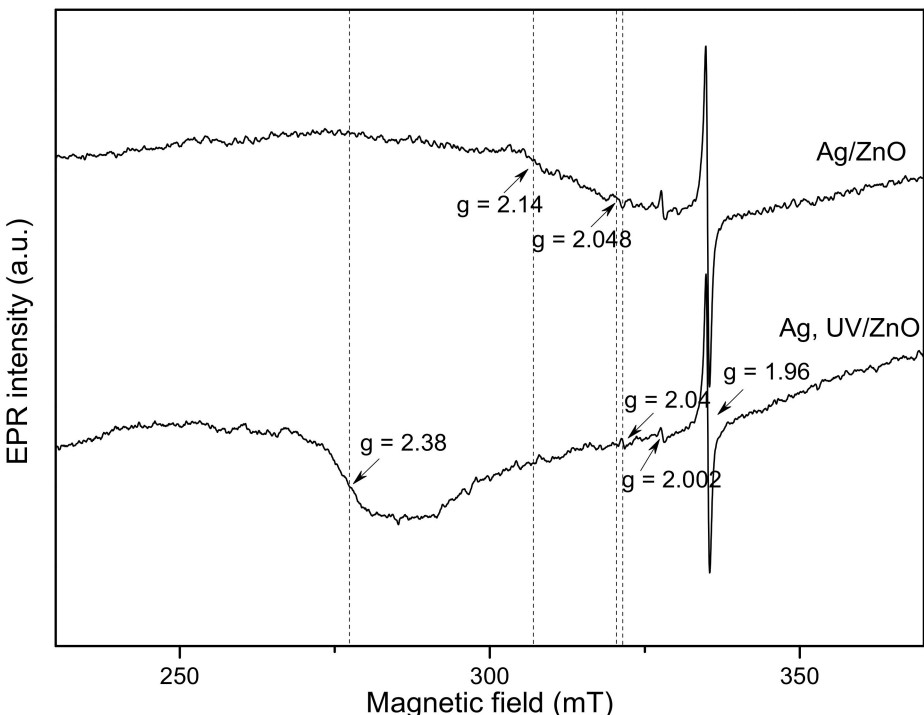

**Figure 4.** EPR spectra of Ag/ZnO and Ag-UV/ZnO films.

Furthermore, differences in the EPR spectra between Ag/ZnO and Ag-UV/ZnO samples are noticeable. In the Ag/ZnO sample, a very weak EPR signal is detected at g = 2.048, potentially associated with absorbed oxygen $O^{2-}$. Moreover, a signal recorded at g = 2.14 in the same sample is probably attributed to Zn vacancy [28].

Meanwhile, in the Ag-UV/ZnO sample, a very weak EPR line with g = 2.04 is detected. Similar lines were observed in previous studies and are associated with defects such as $V_{-Zn}$: $Zn_i^0$ complexes or $V_0^-$ $V_{Zn}$ clusters [28,29]. Another distinguishing observation between the Ag/ZnO and Ag-UV/ZnO samples is the prominent broad signal, recorded at g = 2.38, which is response to the $Ag^{2+}$ center. Previous studies have reported the observation of the $Ag^{2+}$ signal only at deposition temperatures 350 °C, 400 °C and 450 °C and the

$Ag^{2+}$ centers decay within the temperature range of 290–380 °C [29,30]. In our work, the increase in $Ag^{2+}$ centers is linked to an increase in hole concentrations [30]. $Ag^+$ ions, on the other hand, do not manifest the EPR spectrum. These ions are diamagnetic with a $4d^{10}$ electronic configuration. However, under UV illumination, $Ag^+$ ions cans lose $e^-$ and form paramagnetic silver ions ($Ag^{2+}$) with $4d^9$ configurations [30]. When $Ag^+$ ions gain an electron after UV treatment, paramagnetic $Ag^0$ centers with the $4d^{10} 5s^1$ electronic configuration are formed, and their reported g-value is close to that of the free electron. However, $Ag^0$ atoms are not stable and under appropriate annealing or UV irradiation, they may undergo electron excitation, resulting in the formation of $Ag^+$ ions [30].

### 2.2. Optical Characterization

Figure 5a represents the UV–Vis optical absorbance spectra of three samples—pure ZnO and the silver-modified samples without the UV illumination aid (Ag/ZnO) and with the UV illumination aid (Ag-UV/ZnO). The spectra exhibited peaks at wavelengths of 361, 363 and 368 nm for ZnO, ZnO/Ag, and Ag-UV/ZnO, respectively.

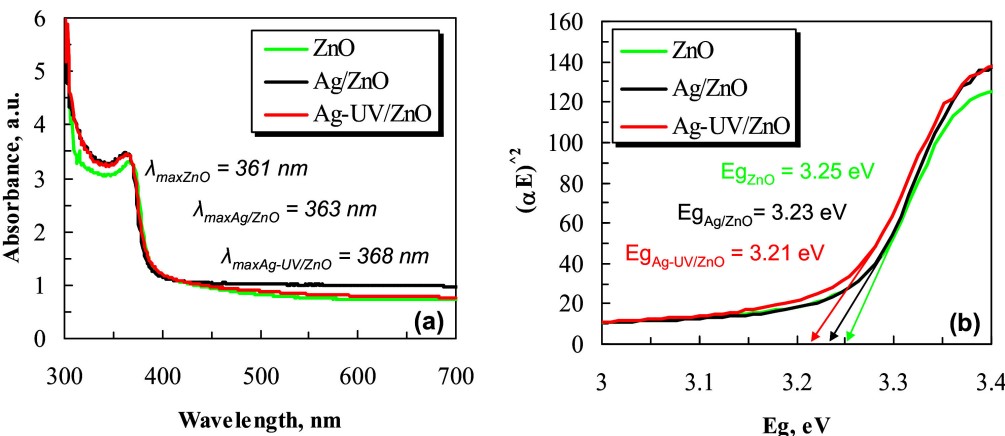

**Figure 5.** (**a**) UV–Vis absorbance spectra and (**b**) optical bandgap energies of ZnO, Ag/ZnO and Ag-UV/ZnO films.

These absorbance peaks correspond to the excitonic peaks observed in UV–Vis absorbance spectra, in accordance with the characteristics of zinc oxide [31]. The shifts of the absorbance peaks towards longer wavelengths are attributed to the interaction between Ag and ZnO [32]. The strong interfacial electronic coupling between zinc oxide and silver ions contributes to a reduction in the loss of light energy [33].

Furthermore, the UV–Vis spectroscopy data were also used to calculate the bandgap energy of the films. The bandgap energy values are obtained by plotting $(\alpha E)^2$ relative to E and extrapolating as a straight line along the *x*-axis using the Kubelka–Munk equation. Figure 5b shows the change in the absorption edge resulting from the inclusion of silver ions in the zinc oxide films. The bandgap energy tends to decrease when the pure films are modified with $Ag^+$. These values closely align with the bandgap energy of zinc oxide [34,35], which plays an essential role in the photocatalytic process. While pure ZnO exhibits a bandgap energy of 3.25 eV, the $E_g$ of silver co-catalytic-modified films decreases to 3.23 eV (Ag/ZnO) and 3.21 eV (Ag-UV/ZnO), as shown in Figure 5b. The silver co-catalytically modified ZnO films prepared with the UV illumination aid (Ag-UV/ZnO) are capable of trapping a higher number of electrons and are thus expected to be more efficient in the oxidation and reduction reactions occurring on the catalyst. Consequently, $Ag^+$ increases the lifetime of photogenerated $e^-/h^+$, effectively limiting their recombination rate. Thus, solely based on the UV–Vis analysis, Ag-UV/ZnO films are expected to exhibit the highest photocatalytic efficiency, while the pure ZnO films are anticipated to have lower activity.

### 2.3. Photocatalytic Efficiency

Photocatalytic tests are conducted to assess the influence of silver modification on the zinc oxide films in the photocatalytic decolorization of organic dye in aqueous solution. The reaction kinetics is interpreted using pseudo-first order reactions [36]. Figure 6 displays the plot of $-Ln(C/Co)$ for the degradation of organic dye by pure and silver co-catalytic modified zinc oxide films, where the $Ag^+$ concentration is varied, comparing Ag-UV/ZnO (Figure 6a) and Ag/ZnO (Figure 6b). The rate constants are consistently higher values in the case of Ag-UV/ZnO, as indicated by a significant increase in the $-Ln(C/Co)$ curve (Figure 6a). Furthermore, it is evidently shown that the photocatalytic efficiency for Ag-modified films increases proportionally to the Ag+ ions concentration during the co-catalyst fixation treatment.

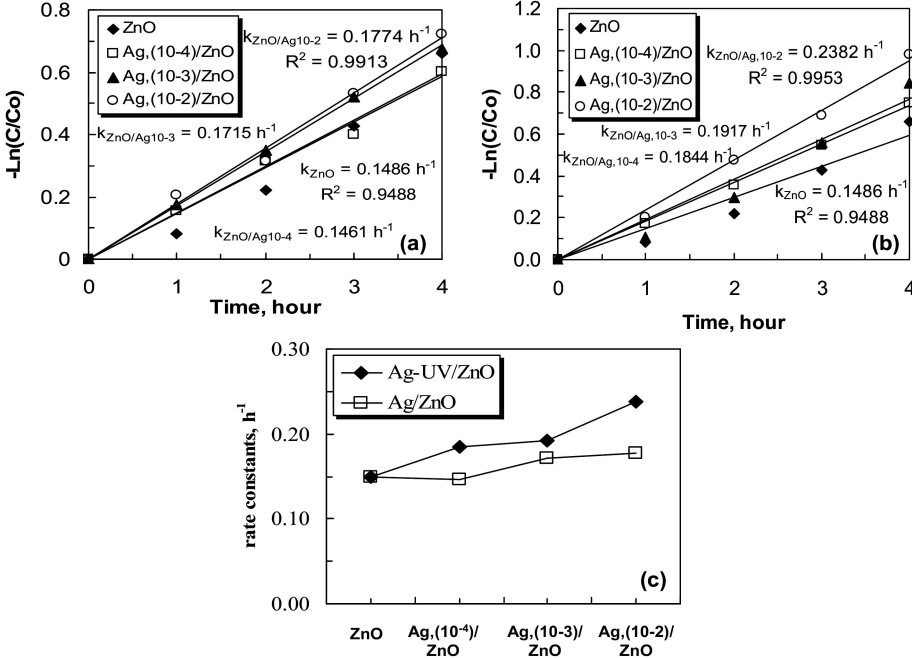

**Figure 6.** Decolorization kinetics of methylene blue using ZnO, Ag/ZnO (**a**) and Ag-UV/ZnO (**b**). The rate constants from photocatalytic processes of sol–gel films modified with silver ions: $10^{-2}$, $10^{-3}$, $10^{-4}$ M (**c**).

The graph of $-Ln(C/Co)$ vs. at time demonstrates that Ag/ZnO, prepared at $Ag^+$ ions concentration of $10^{-2}$ M, has the highest photocatalytic activity, while the remaining samples show a decrease in efficiency. This observation is supported by the comparison of the rate constants of the pure ZnO and Ag($10^{-4}$ M)/ZnO, Ag($10^{-3}$ M)/ZnO, Ag($10^{-2}$ M)/ZnO films, which are 0.1486 $h^{-1}$, 0.1461 $h^{-1}$, 0.1715 $h^{-1}$ and 0.1774 $h^{-1}$, respectively (Figure 6c). The higher values of the rate constants (Figure 6c) correspond to a faster degradation of the dye, as UV light can excite electrons in the catalyst and thereby oxidize the organic pollutant. This observation can also be related to a reduction of the recombination rate of recombination of the formed charge pairs ($e^-/h^+$) and the narrowing of the bandgap width in the Ag-modified samples. The bandgap energies of the three types of nanostructure films are investigated and presented in Figure 5, which shows a decrease in the $E_g$ due to the Ag modification of the films. However, the same observation cannot be made for the films treated without UV illumination aid (Ag/ZnO), in comparison to which the pure, unmodified ZnO catalyst still has the highest photocatalytic efficiency (Figure 6b). Further investigations are needed to provide additional evidence and a better understanding of this discrepancy by elucidating the detrimental effects of $Ag^+$ impregnation treatment on the photocatalytic activity of the pure ZnO samples.

The photocatalytic degradation of methylene blue using the three different photocatalyst samples is shown in Figure 7. After 4 h of stirring under UV light illumination, the Ag-modified nanostructure films, treated photo-fixed without UV irradiation (Ag/ZnO), exhibit a lower degradation rate compared to the pure zinc oxide (Figure 7a). This observation can be attributed to the fact that the co-catalyst does not undergo further excitation, leading to the blocking of the ZnO surface active centers. However, in the case of Ag-UV/ZnO films, a linear increase in efficiency was observed with an increase in the Ag co-catalyst concentration during the treatment step (Figure 7b). Therefore, the silver ions photo-fixation products act as highly efficient electron traps and improve the Ag/ZnO couple as a catalyst for methylene blue degradation.

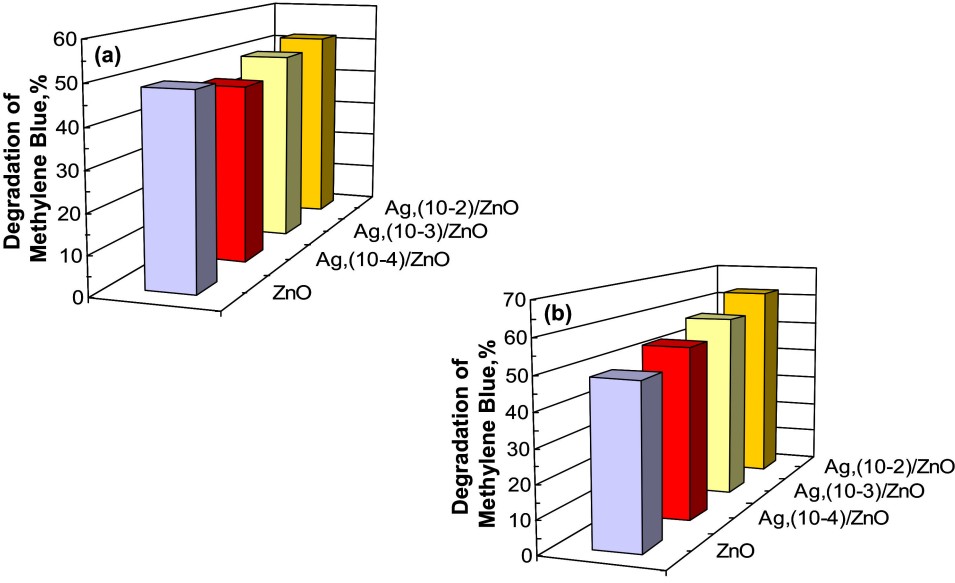

**Figure 7.** Percentage of methylene blue dye degradation using pure and silver ZnO photo-fixed without (**a**) and with (**b**) UV illumination.

The results obtained from the Total Organic Carbon (TOC) analysis (Figure 8) indicate that the pure ZnO films have a lower percentage of MB mineralization in comparison with silver-modified ones. In the case of the Ag-UV/ZnO catalyst, the highest percentage of TOC of mineralization of the dye is observed with the respective values of 40.73% for ZnO, 42.57% for Ag/ZnO, 45.23% for Ag-UV/ZnO. Furthermore, the TOC values for methylene blue are consistently lower than the ones obtained via UV–Vis spectroscopy during the photocatalytic experiments, which can be attributed to the formation of non-absorbing intermediate products that cannot be registered in the UV–Vis absorbance spectra.

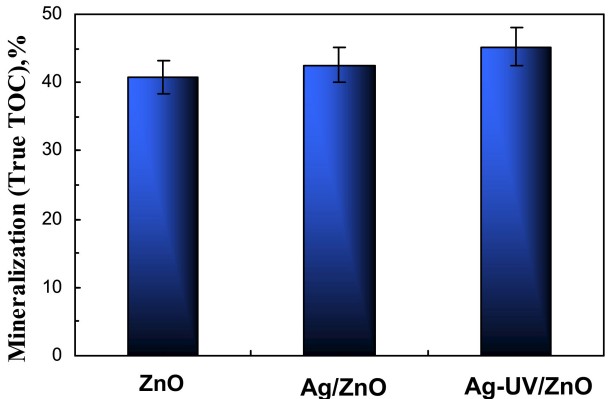

**Figure 8.** Histograms of TOC conversion after photocatalysis—fourth hour of UV illumination.

Such intermediates, for example, were reported by Nayak et al. during the photocatalytic decolorization of methylene blue [36] and were found to exhibit a lower interaction with the hydroxyl radicals generated during photocatalysis, effectively accumulating in the process. Consequently, the TOC analysis can be shown to complement the analysis of the complex photodegradation process, revealing the true rate of degradation of the intermediates and thus complementing the UV–Vis measurements, typically used to obtain liquid-phase photodegradation of dyes.

Figure 9 shows the concentration profiles of methylene blue removal by ZnO and Ag $(10^{-2}$ M)/ZnO at different types of light illumination. The dye spectrum decreases with the time of illumination. As expected, the concentration of the dye decreases most rapidly under UV light, followed by the visible, and finally in darkness, where only adsorption of dye molecules is expected to occur. Furthermore, for the silver-modified catalysts, UV light illumination also leads to a decrease in the recombination level of the photogenerated charges. Upon irradiation with visible light, silver ions are unable to significantly prevent the loss of quantum yield.

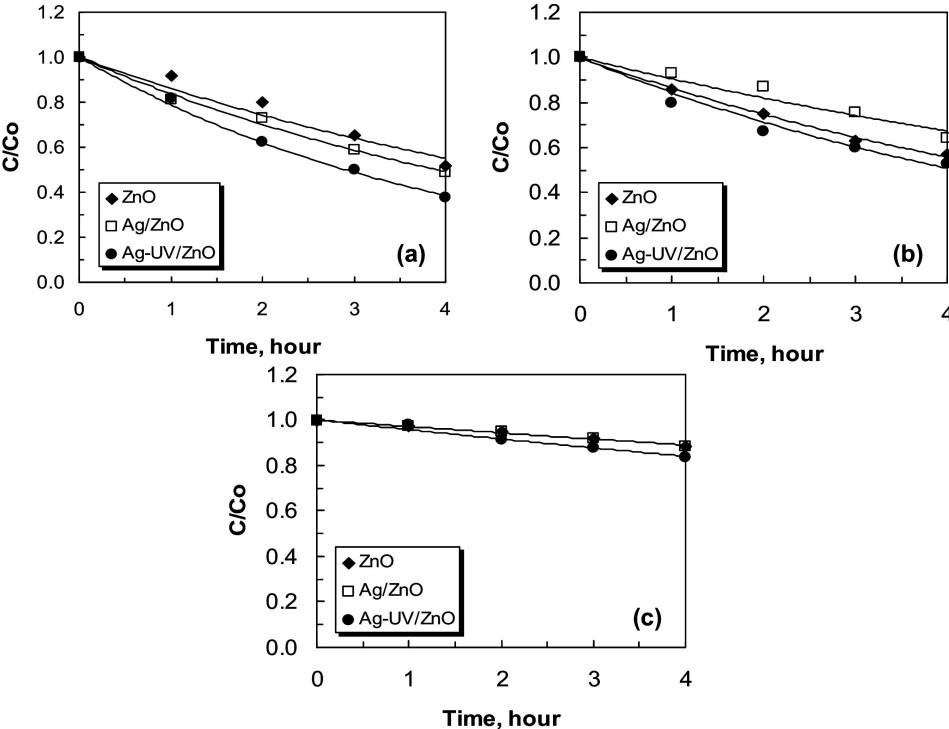

**Figure 9.** Concentration spectra of dye using pure and silver $(10^{-2}$ M)-modified ZnO films photofixed without and with ultraviolet illumination. Photocatalytic tests are carried out in the presence of UV (**a**), visible light (**b**) and in darkness (**c**).

Moreover, this data once again confirm that the silver-modified catalyst exhibits the highest dye removal rate, suggesting limited recombination of the photogenerated charges. Since visible light is not expected to lead to charge-pair generation, in this case, the treatment with silver ions is not beneficial since they are unable to significantly prevent the loss of quantum yield. This control experiment also shows that light with insufficient energy on its own is not able to lead to the photodegradation of dye, as well as that no plasmonic effects of the surface-fixed Ag are observed.

The concentration of methylene blue exhibits a decrease in darkness, albeit at a lower rate, compared to the photocatalytic process under light illumination since it is governed by adsorption. This is visible in Figure 9c, showing a constant value of methylene blue dye concentration decrease in the case of pure ZnO and Ag/ZnO samples.

## 3. Materials and Methods

Sol–gel zinc oxide sol–gel films were deposited on glass slide substrates (ca. 76 mm × 26 mm, ISO-LAB, Schweitenkirchen, Germany) using the dip-coating method. The sol–gel solution was prepared by mixing zinc acetate dehydrate ($Zn(CH_3COO)_2 \cdot 2H_2O$, Fluka, >99.5%), 2-methoxyethanol ($C_3H_8O_2$, Fluka, >99.5%), monoethanolamine ($C_2H_7NO$, Fluka, >99.5%) as precursor, solvent and a stabilizer, respectively, mixed in 1:4:1 molar ratio. The solution was stirred at room temperature for 15 min, followed by heating to 60 °C for 1 h to obtain clear and homogenous sol. The sol–gel films were deposited with five coating cycles, and each immersion was followed by drying at 100 °C for 1 h to remove the solvent. The films were then annealed at 500 °C for one hour to improve the crystalline structure.

The silver cocatalyst modification was achieved by a simple chemical photodeposition method. The pure films were modified with silver nitrate ($AgNO_3$, Pub-Chem, United States) solution ($10^{-2}$–$10^{-4}$ M) for 20 min, either in darkness or under UV light illumination. Afterward, the modified films were rinsed with distilled water and dried at 100 °C for ten minutes (Figure S1).

The surface morphology of the sol–gel films was observed using scanning electron microscopy (SEM), Hitachi TM4000 (accelerating voltage 15 kV, Krefeld, Germany) was applied to observe the surface morphology of sol–gel films. Energy-dispersive X-ray (EDX) equipment (Bruker AXS Microanalysis GmbH, Berlin, Germany) was performed, employing a Bruker AXS detector. The UV–Vis spectrometer model, Evolution 300 Thermo Scientific (Madison, WI, USA), was used to evaluate the optical absorption of materials.

The crystal parameters of the obtained samples were estimated by an X-ray diffractometer (Siemens D500 with CuK$\alpha$ radiation within 2$\theta$ range 20–80° at step of 0.05° 2$\theta$ and counting time 2 s/step, Karlsruhe, Germany) for phase and structure identification. The crystallite sizes were assessed using the Scherrer equation.

X-ray photoelectron measurements were carried out on the ESCALAB MkII (VG Scientific, now Thermo Scientific, Manchester, UK) electron spectrometer with a base pressure in the analysis chamber is of $5 \times 10^{-10}$ mbar, provided with twin anode MgK$\alpha$/AlK$\alpha$ non-monochromated X-ray source used excitation energies of 1253.6 and 1486.6 eV, respectively. The measurements were provided only with AlK$\alpha$ non-monochromated X-ray source. The instrumental resolution is about 1 eV. The data were analyzed by SpecsLab2 CasaXPS software (2.3.25PR1). Because of electrostatic sample charging the energy scale has been calibrated by normalizing the Zn2p to 1022.0 eV for the zinc-containing sample, respectively. The processing of the measured spectra includes a subtraction of X-ray satellites and Shirley-type background. The peak positions and areas were evaluated by a symmetrical Gaussian–Lorentzian curve fitting.

EPR measurements were carried out on a JEOL JES-FA 100 EPR spectrometer (Tokyo, Japan) provided with a standard TE011 cylindrical resonator. The EPR spectra were recorded at room temperature in the following conditions: modulation frequency 100 kHz, microwave power 1.26 mW, modulation amplitude 0.2 mT, time constant 0.1 s and sweep time 2 min.

The photocatalytic degradation of organic pollutant was investigated at 23 °C in photoreactor (200 mL volume) under ultraviolet (Sylvania BLB18 W, 315–400 nm of emission range) and visible light irradiation and in darkness. A methylene blue ($C_{16}H_{18}ClN_3C$, Merck, Rahway, NJ, USA) solution with a concentration of 5 ppm was used a model contaminant. The first fifteen minutes of the process are carried out in complete darkness to allow adsorption–desorption. Then the light is turned on. During the photocatalytic tests, aliquots (2 mL) are taken at regular intervals in order to monitor the activity of the catalyst and measured, in terms of absorbance of the remaining dye using a UV–Vis spectrum analyzer at 666 nm (maximum absorption wavelength). The degradation of methylene blue D (%) was calculated using Equation (1):

$$D(\%) = \frac{C_0 - C}{C_0} * 100, \tag{1}$$

where $C_0$ is the initial concentration, and C is the concentration of methylene blue after light irradiation.

The photo-stability of pure ZnO, Ag/ZnO and Ag-UV/ZnO films is investigated, as the photocatalytic tests were repeated three times. Each experiment was conducted with a new dye solution with a concentration of 5 ppm. All the photocatalytic data were reproducible, which led to our conclusion about the stability of nanostructures.

The total organic carbon (TOC) was investigated for the treated methylene blue solutions, using the high-temperature (850 °C) catalytic oxidation method and Elementar Vario Select TOC analyzer. The standard variation was estimated based on three measurements of each sample. The percentage of mineralization of the methylene blue was evaluated using Equation (2):

$$Mineralization(\%) = \frac{TOC_0 - TOC}{TOC_0} * 100, \tag{2}$$

where $TOC_0$ is the initial TOC of the dye solution, and TOC is the final TOC at t time.

## 4. Conclusions

In conclusion, the present study presents a novel method of modifying photocatalytic ZnO thin films by impregnation–fixation and photo-fixation of silver ions and thus investigates the influence of UV illumination aid on this process. The results demonstrate that this method is cost-effective, simple, and yields excellent photocatalytic characteristics of the UV-treated thin films. The optics and photocatalytic properties the significant contribution of $Ag^+$ ions in electron-trapping properties, and demonstrate that the UV photo-fixation enhances the interaction and charge transfer in ZnO, reducing the $e^-/h^+$ pairs recombination. Therefore, silver co-catalytically zinc oxide films photo-fixed with UV have the highest photocatalytic activity. The development of an efficiently modified photocatalyst for visible light applications represents a potential extension of the applications of these semiconductor materials.

**Supplementary Materials:** The following supporting information can be downloaded at: https://www.mdpi.com/article/10.3390/catal13071121/s1, Figure S1: Procedure for obtaining of ZnO and Ag/ZnO/Ag sol-gel films.

**Author Contributions:** Conceptualization, N.K.; methodology, N.K. and D.I.; photocatalytic experiments, D.I.; data curation, D.I., R.M., H.K. and N.K.; writing—original draft preparation, N.K.; writing—photocatalysis D.I and N.K. All authors have read and agreed to the published version of the manuscript.

**Funding:** This research was funded by the Bulgarian NSF project KP-06-N59/11 (КП-06-Н59/11).

**Data Availability Statement:** Not applicable.

**Acknowledgments:** The authors are grateful to financially supported by the Bulgarian NSF project KP-06-N59/11 (КП-06-Н59/11).

**Conflicts of Interest:** The authors declare no conflict of interest.

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
