# Peer review of "Effect of Ultraviolet Illumination on the Fixation of Silver Ions on Zinc Oxide Films and Their Photocatalytic Efficiency"

_catalysts, doi:10.3390/catal13071121_

Round 1

Reviewer 1 Report

The manuscript reported the synthesis of Ag/ZnO films for photocatalytic degradation of MB. It could be accepted after major revision. The following issues need to be addressed before the manuscript could be accepted:

1.       The Title of the manuscript should be revised to show the catalytic application of this material.

2.       The synthesized material was Ag-doped ZnO films. Thus, please change the “ZnO/Ag UV films” to “Ag/ZnO films”.

3.       What is the valance state of Ag in the Ag-doped ZnO films? Is it zero valent Ag(0) or one valent Ag+? If this is Ag+, the Ag should be existed as oxides (Ag2O), and the film should be Ag2O/ZnO.

4.       HRTEM showing the crystal lattice of Ag is suggested to be conducted to confirm the Ag(0) or Ag2O.

5.       How about the photo-stability of the Ag/ZnO films? Recycling experiments could be conducted.

6.       How about the possible release of Ag+ during the photocatalytic experiments? ICP could be used to quantify the release concentration of Ag+.

7.       The resolution of the Figures are relatively low. Please enhance the resolution and quality of the Figures.

8.       Figure 10 and Figure 11 can be put in Supporting Information.

The English should be polished before acceptance.

Author Response

Response to Reviewer 1

Ms. Ref. No.: 2475885

Title: “Effect of Ultraviolet Illumination on the Fixation of Silver Ions on Zinc Oxide Films and their Photocatalytic Efficiency”, written by Dobrina Ivanova, Ralitsa Mladenova, Hristo Kolev, Nina Kaneva

            Catalysts

We thank very much to the Reviewer for the valuable marks, comments and suggestions, supporting the presentation of our work. We accept all the recommendations and corrected our paper properly following the suggestions given by the Reviewer in the same sequence.

Hope you find revised manuscript suitable for publication in Catalysts. The English language has been corrected throughout the article. Corrections are reflected in blue. We look forward to hearing from you in due course.

Reviewer 1: The manuscript reported the synthesis of Ag/ZnO films for photocatalytic degradation of MB. It could be accepted after major revision. The following issues need to be addressed before the manuscript could be accepted:

  1. The Title of the manuscript should be revised to show the catalytic application of this material.

The title of the article is changed – catalytic application present: “Effect of Ultraviolet Illumination on the Fixation of Silver Ions on Zinc Oxide Films and their Photocatalytic Efficiency”

  1. The synthesized material was Ag-doped ZnO films. Thus, please change the “ZnO/Ag UV films” to “Ag/ZnO films”.

We everywhere in the article (text and all figures) are changed ZnO/Ag to Ag/ZnO, as recommended.

  1. What is the valance state of Ag in the Ag-doped ZnO films? Is it zero valent Ag(0) or one valent Ag+? If this is Ag+, the Ag should be existed as oxides (Ag2O), and the film should be Ag2O/ZnO.

We modified the pure films with silver nitrate, ie. Ag+. The XPS method confirms the oxidation state of silver (+1) - we have described it in the article. There are oxygen ions in the sample (ZnO), but there is no way to prove Ag2O. For this reason we write only Ag+.

  1. HRTEM showing the crystal lattice of Ag is suggested to be conducted to confirm the Ag(0) or Ag2O

We would like to thank the reviewer for the question and recommendation.

But in our study we used thin films. For this reason, we cannot use TEM to characterize the samples. Therefore, we characterized the samples with XPS and EPR. In future studies, we will synthesize a powder, which we will again modify with silver - then we will be able to use TEM (the beam will be able to penetrate) and find out if it is Ag2O.

  1. How about the photo-stability of the Ag/ZnO films? Recycling experiments could be conducted?

We investigated the stability of the materials as recommended.

Discussion is added to the text: “The photo-stability of ZnO, Ag/ZnO and Ag-UV/ZnO films is investigated, as the photocatalytic tests were repeated three times. Each experiment was conducted with a new dye solution with a concentration of 5 ppm. All the photocatalytic data were reproducible, which made us to conclude about the stability of nanostructures.” P11, lines 380-383

We would like to thank the reviewer for the question with recycling. If, it implies if the re-cycling stability of the photocatalyst is tested in this work - we must answer no, since it is outside the scope of the study. The purpose of these experiments was to investigate the optimal concentration of the co-catalyst to achieve enhancement of the photocatalytic activity. Follow up works, investigating the stability and properties of the best-case-scenario of the modification will be carried promptly.

  1. How about the possible release of Ag+ during the photocatalytic experiments? ICP could be used to quantify the release concentration of Ag+.

In our other research work (TiO2/Ag powder films), we have used the ICP method to track the presence of silver in the solution of the dye after photocatalysis. Experimental results show that there are no silver ions in the solution. The zinc oxide system with silver is also yet to be examined. We assume that the end results will be the same.

  1. The resolution of the Figures are relatively low. Please enhance the resolution and quality of the Figures.

We have improved the quality and resolution of all figures, as recommended.

  1. Figure 10 and Figure 11 can be put in Supporting Information.

We put them in Supporting Information (now Figure 10 is Figure S1, Figure 11 is Figure S2), as recommended.

Reviewer 2 Report

The article may require a major revision before being published on the Catalysts due to the data quality and lack of scientific presentation. Here are some suggestions that may not represent all similar issues.

1. In the abstract, I suggest the authors put the quantitive description instead of pure interpretation-based language, such as " generally higher photocatalytic activity" "higher photocatalytic efficiency" and "exhibit substantial activity".

2. Page 2, line 54, CdS is not environmentally friendly. Bulk TiO2 is an insulator while nanoparticles can be semiconductors. Suggest rewriting.

3. Page 2, lines 64-67, from Citation 16, it is the Ag nanoparticles that can be an electron acceptor or electron trap, not silver. 

4. Figure 1, there isn't any direct evidence indicating the wrinkling is ZnO. EDX mapping image may help. 

5. Figure 5, in UV-vis, I can't tell the difference between ZnO/Ag and ZnO/AgUV. And ZnO absorption peak seems to have a longer wavelength. Need more detailed data/plot.

Author Response

Response to Reviewer 2

Ms. Ref. No.: 2475885

Title: “Effect of Ultraviolet Illumination on the Fixation of Silver Ions on Zinc Oxide Films and their Photocatalytic Efficiency”, written by Dobrina Ivanova, Ralitsa Mladenova, Hristo Kolev, Nina Kaneva

            Catalysts

We thank very much to the Reviewer for the valuable marks, comments and suggestions, supporting the presentation of our work. We accept all the recommendations and corrected our paper properly following the suggestions given by the Reviewer in the same sequence.

Hope you find revised manuscript suitable for publication in Catalysts. The English language has been corrected throughout the article. Corrections are reflected in blue. We look forward to hearing from you in due course.

Reviewer 2: The article may require a major revision before being published on the Catalysts due to the data quality and lack of scientific presentation. Here are some suggestions that may not represent all similar issues:

  1. In the abstract, I suggest the authors put the quantitive description instead of pure interpretation-based language, such as "generally higher photocatalytic activity" "higher photocatalytic efficiency" and "exhibit substantial activity".

We added the quantitative description instead words "generally higher photocatalytic activity", "higher photocatalytic efficiency" and "exhibit substantial activity", as recommended.

Discussion is added to the text: “It is observed that the photocatalytic activity increases proportionally to the Ag+ ion concentration: ZnO < Ag(10-4 M)/ZnO, < Ag(10-3 M)/ZnO < Ag(10-2 M)/ZnO. Moreover, the catalysts modified under UV illumination during the fixation treatment (Ag-UV/ZnO) exhibited a higher photocatalytic efficiency and degradation the dye in comparison with those without a light source (Ag/ZnO). The experimental results are confirmed using total organic carbon (TOC) analysis. The optimal silver concentration (10-2 M) is established, which shows the highest photocatalytic efficiency (in both cases of ion fixation treatment).”

  1. Page2, line 54, CdS is not environmentally friendly. Bulk TiO2 is an insulator while nanoparticles can be semiconductors. Suggest rewriting

Yes, CdS is not environmentally. We deleted CdS and added nanostructured semiconductors: “Photocatalysts offer the advantage of enhancing the efficiency of the wastewater treatment process by allowing both spontaneous and non-spontaneous reactions to take place [10], and to effectively degrade organic dyes into simpler products, that pose lower environmental risk. Among the popular photocatalysts, nanostructured semiconductors, such as TiO2, ZnO and Fe2O3 are preferred due to their availability, cost-effectiveness, and environmental compatibility [11].” P.2, lines 46-52

  1. Page 2, lines 64-67, from Citation 16, it is the Ag nanoparticles that can be an electron acceptor or electron trap, not silver.

We added to the text: “Among these, silver stands out as one of the most employed due to its excellent electrical conductivity, chemical stability and high reduction potential. When introduced onto the ZnO surface, Ag nanoparticles can act as еlеctrоn traps or electron acceptors.” P.2, line 61-64

  1. Figure 1, there isn't any direct evidence indicating the wrinkling is ZnO. EDX mapping image may help

We used SEM analysis to determine the surface morphology of the thin films. Additionally, we wanted to understand whether silver modification affects the surface. For the presence of zinc, oxygen and silver, we use EDX.

  1. Figure 5, in UV-vis, I can't tell the difference between ZnO/Ag and ZnO/AgUV. And ZnO absorption peak seems to have a longer wavelength. Need more detailed data/plot

We further measured the optical absorption spectra of the three samples, as recommended.

Hopefully, the figure is of better quality. The energies of the band gap in the films are very close indeed.

Round 2

Reviewer 1 Report

The manuscript can be accepted in the current form. 

Author Response

Thank you very much!

Reviewer 2 Report

The authors made some changes regarding the manuscript. However, I don't see the difference or effort. In my previous view, number 5, in Figure 5a, the data quality has not been improved and the data is against the claim. Lambda maxZnO = 361 nm has a longer wavelength than Lambda max ZnO/Ag UV = 368 nm. The authors indicate, "We further measured the optical absorption spectra of the three samples, as recommended." Please be specific about what has been improved or changed to address the obvious contradiction between the plot and claim. 

Author Response

Dear Reviewer, We have changed the quality and clarity of Fig. 5b.
We did remeasure the absorption spectra of the three types of films - the results show that the lambda max of the pure zinc oxide (361nm) is no longer than the lambda max of the silver modified ones (365, 368 nm).
This is what we wrote in our article - Page 6, Line 211-214 "The shifts of the absorbance peaks towards longer wavelengths are attributed to the interaction between Ag and ZnO. The strong interfacial electronic coupling between zinc oxide and silver ions contributes to a reduction in the loss of light energy".
Other authors also reach this conclusion -
Singh, R.; Barman, P.B.; Sharma, D. Synthesis, structural and optical properties of Ag doped ZnO nanoparticles with enhanced photocatalytic properties by photo degradation of organic dyes. Mater. Sci. Mater. Electron. 2017, 28, 5705–5717.
R. Rajendran, A. Mani, Photocatalytic, antibacterial and anticancer activity of silver-doped zinc oxide nanoparticles, Journal of Saudi Chemical Society, 2020, 24, 1010-1024.